# Occupational Tuberculosis Among Laboratory Workers in South Africa: Applying a Surveillance System to Strengthen Prevention and Control

**DOI:** 10.3390/ijerph17051462

**Published:** 2020-02-25

**Authors:** Jennica Garnett, David Jones, Graham Chin, Jerry M. Spiegel, Annalee Yassi, Nisha Naicker

**Affiliations:** 1School of Population and Public Health, University of British Columbia, Vancouver, BC V6T 1Z4, Canada; jennica.garnett@alumni.ubc.ca (J.G.); jerry.spiegel@ubc.ca (J.M.S.); annalee.yassi@ubc.ca (A.Y.); 2Safety, Health, Environment Department, National Institute for Occupational Health (NIOH), a division of National Health Laboratory Service (NHLS), Johannesburg 2000, South Africa; DavidJ@nioh.ac.za (D.J.); GrahamC@nioh.ac.za (G.C.); 3Epidemiology and Surveillance Section, National Institute for Occupational Health (NIOH), a division of National Health Laboratory Service (NHLS), Johannesburg 2000, South Africa; 4School of Public Health, University of the Witwatersrand, Johannesburg 2000, South Africa; 5Department of Environmental Health, Faculty of Health Science, University of Johannesburg, Johannesburg 2000, South Africa

**Keywords:** tuberculosis, occupational health, occupational health surveillance, laboratory workers, healthcare workers

## Abstract

Background: Tuberculosis (TB) is recognized as an important health risk for health workers, however, the absence of occupational health surveillance has created knowledge gaps regarding occupational infection rates and contributing factors. This study aimed to determine the rates and contributing factors of active TB cases in laboratory healthcare employees at the National Health Laboratory Service (NHLS) in South Africa, as identified from an occupational surveillance system. Methods: TB cases were reported on the Occupational Health and Safety Information System (OHASIS), which recorded data on occupation type and activities and factors leading to confirmed TB. Data collected from 2012 to 2019 were used to calculate and compare TB risks within NHLS occupational groups. Results: During the study period, there were 92 cases of TB identified in the OHASIS database. General workers, rather than skilled and unskilled laboratory workers and medical staff, had the highest incidence rate (422 per 100,000 person-years). OHASIS data revealed subgroups that seemed to be well protected, while pointing to exposure situations that beckoned policy development, as well as identified subgroups of workers for whom better training is warranted. Conclusions: Functional occupational health surveillance systems can identify subgroups most at risk as well as areas of programme success and areas where increased support is needed, helping to target and monitor policy and procedure modification and training needs.

## 1. Introduction

Despite the United Nations’ 2018 High Level meeting on the “Fight Against TB” that recognized healthcare workers as a high-risk group, accurate and timely information on occupational exposures remains limited, which puts these workers in jeopardy. Numerous studies have concluded that health workers are subject to increased rates of infection which are significantly higher than the general population, with particular concern for low- and middle-income settings in high tuberculosis (TB) incidence countries [1,2,3,4,5,6]. However, there is a lack of research investigating the differentiation in exposure and disease between the subcategories of healthcare workers, such as laboratory workers, who may have a significantly different education and interaction with healthcare delivery. It has been documented that there is likely a three to six-fold increased risk of TB in healthcare workers; this information is generalized and does not provide specific details regarding the methods of transmission or effective ways to interrupt the cycle of infection [7]. Very few studies of laboratory-acquired occupational TB infection have been undertaken; most reports are of small outbreaks in specific laboratories based in a low burden, high income country [7]. 

South Africa has one of the highest rates of TB in the world, 520 per 100,000 in 2018, and faces many challenges in the collective effort to decrease the negative impact and prevalence of the disease [8]. This is complicated by the high prevalence of human immunodeficiency virus (HIV), which has created a population that is particularly vulnerable to TB, and the increasing presence of multi-drug resistant TB [3,8]. Similar to other low- to middle-income countries (LMIC) with high TB transmission, South Africa’s public healthcare system faces challenges such as overcrowding and limited resources, which have decreased accessibility to TB diagnosis and treatment [3]. While one of the greatest limitations in health systems is the shortage of healthcare workers, difficult working conditions and a high burden of infectious disease have been observed to be some of the barriers to retaining already scarce human resources in this sector [9]. Continued outbreaks and TB transmissions associated with frequent unprotected exposure to TB patients or bacteria cause significant concern in the healthcare sector, as outbreaks cause increased stress of workers, increased absences from work and additional financial burdens on already limited budgets [10]. The persistence of TB as a notable occupational hazard is further aggravated by weak occupational health programmes, limited healthcare resources, and reluctance to disclose TB diagnosis due to stigma [6,7]. However, the extent and nature of the problem remains obscured in the absence of surveillance.

Optimising the performance, quality and retention of a healthy workforce through evidence-informed policies provides a way to strengthen the health system for both staff and patients [11]. In this regard, an effective information surveillance system can estimate infection rates; characterize infection contributors; design, implement and evaluate preventive programmes; improve knowledge of disease transmission; influence policy; and identify research needs. Previous literature has largely focused on the transmission factors of those directly caring for TB patients, leaving a general underrepresentation of adjunct healthcare workers including laboratory workers and general employees. There is an urgent need for effective surveillance programmes for occupational TB that have the ability to evaluate factors of exposure specific to the occupational groups within the healthcare system. Such a surveillance system could provide a better understanding of the distribution of TB and the factors that aid or hinder transmission in a timely and comprehensive manner, as well as be used to enhance prevention strategies and support future policy and procedures [5,6,11,12,13]. Interventions informed by surveillance evidence allow for prevention, early diagnosis, education and training of healthcare workers, and may be effective in decreasing prevalence in a notably vulnerable group [1]. 

Through a partnership of two World Health Organization (WHO) collaborating centres engaged in promoting the occupational health of health care workers, the Occupational Health and Safety Information System (OHASIS) was developed to meet the challenge posed by inadequate available information [13,14]. As laboratory workers constitute one of the groups of health workers observed to be at risk of bacterial exposure and infection [15], the OHASIS system was introduced at the National Health Laboratory Service (NHLS) in 2011. This study seeks to investigate the insights into the incidence of occupational TB and prevalence of risk factors that can be gained from applying a comprehensive surveillance approach. In examining the specific risks of TB faced by laboratory workers, it aims to extend literature that has previously been largely devoted to risks in high-income low-TB incidence settings, and to consider circumstances in a high-risk national environment.

## 2. Materials and Methods

### 2.1. Setting and Sample

In 2011, the National Institute of Occupational Health (NIOH) in South Africa, in collaboration with the University of British Columbia, initiated the implementation of an information system to support the occupational health and safety of laboratory workers, with focused attention on monitoring occupational TB. OHASIS is a comprehensive set of modules that facilitates the reporting and investigation of occupational exposures and injuries with a specific module dedicated to TB exposures and cases. NIOH implemented the OHASIS system at the affiliated NHLS sites as a paper-based system and then in 2012 as a web-based system. Previously, NHLS TB reporting required contacting the occupational health and safety staff who would then record the incident. However, this type of ad hoc paper-based reporting has multiple challenges such as incomplete data, poor incident reporting, and difficulties reviewing large amounts of data. Moreover, OHASIS provided not only a standardized interface to collect all needed information, but provided an opportunity for staff to self-report. While reporting directly to the occupational health staff is still possible, the revised process entails the staff member reporting directly through OHASIS. The surveillance information system was coupled with strategic education and teaching programmes to enhance occupational TB exposure awareness. This programme is continually supported by occupational health staff and administration and continues to undergo updates and revisions based on feedback. 

### 2.2. Data Collection

As noted above, potential TB exposures were self-reported in OHASIS or recorded in OHASIS by, or with the assistance of, an occupational health nurse (OHN) interviewing the exposed worker. Active medical surveillance was also conducted by the occupational health department who distributed TB screening questionnaires quarterly to all at risk NHLS staff in order to identify employees with potentially active TB. The questionnaire was also distributed to staff upon hire, as well as to affected departments after a confirmed TB case was identified; if deemed necessary, the OHN would follow up to promote completion of the questionnaire. The reporting period used for this study was 16th August 2011 to 16th August 2019. Users were able to backdate their incidents, which allowed for incidents that occurred as early as 1st April 2011 to be included. To accommodate for delayed reporting of TB exposure, employees were able to report incidents of exposure during the observation period up to one month after the end date. 

Cases with missing variables were cross-referenced with human resource files to ensure that all cases had complete employee information. When entering the data in OHASIS, the worker selected all “activities” that related to the potential TB exposure being reported, with selections including: cleaning, office work, sharps handling, material handling, maintenance, motor vehicle accident, operation of equipment, laboratory work, patient handling, transportation of hazardous materials and non-specific (Appendix A). OHASIS then directs the worker to select all that apply with regards to “actions following the incident”, the selections including: no medical attention required, on-site first aid, occupational health unit, outsourced service provider, going to casualty, and private physician (Appendix A). After the individual submits the incident form, it is received by an OHN who then investigates the potential exposure and completes the second half of the form. The OHN selects all “factors that contributed to the potential exposure” under the main headings: environment, worker, patient, work practice, equipment, and administration (Appendix A). Overall, the variables are divided between self-reported activities that led to exposure, the action taken by the employee following the exposure, and the potential factors that led to the exposure as identified by the OHN.

Only cases with a confirmed TB diagnosis were selected from the OHASIS database and analysed for this study. TB was confirmed by any South African approved method of diagnosis. When an employee is diagnosed with TB, it is not compulsory to formally report to the employer or to the government for compensation. Following the direction of the Department of Labour for South Africa, all cases of pulmonary TB are presumed to be work-related if pulmonary TB is transmitted to an employee during the performance of healthcare work from a patient living with active TB or the analysis or testing of infected tissues or fluids (Department of Labour, 2003). Therefore, all cases of TB captured by NHLS are considered occupational diseases. 

### 2.3. Data Analyses

The main study objective was to compare TB rates within occupations represented in the surveillance system, based on potential exposure to diagnosed and undiagnosed TB patients, and the invasiveness of the TB specimen interaction. Four occupational categories were created based on potential occupational exposure to TB: general worker, medical staff, skilled laboratory staff and unskilled laboratory staff. The general worker’s occupation is associated with a laboratory, but the job description does not require regular contact with TB specimens. Occupations in this category include security guard, switchboard assistant, quality assurance officer, cleaner, typist, driver, IT support engineer, and clerk. The member of medical staff’s occupation is associated with a laboratory or clinic and a job description that requires regular contact with potentially active TB patients. Occupations in this category include nurse, phlebotomist, and registrars. The skilled laboratory occupation is associated with a laboratory and a job description that requires regular contact and manipulation of TB specimens. Occupations in this category include medical technician, medical technologist, pathologist, medical scientist, laboratory manager, and laboratory assistant. The unskilled laboratory occupation is associated with a laboratory and a job description that requires regular contact with sealed and unsealed TB specimens. Occupations in this category include laboratory clerk and messenger. All levels of each job were included—from student to supervisor. The general worker has the lowest potential exposure and the skilled laboratory worker has the highest risk of exposure, followed by the unskilled laboratory worker and medical staff. Occupations with inconsistent job descriptions in relation to potential TB exposure were excluded. After the exclusion criteria were applied, 78.04% of the sampled workforce across the 270 laboratory sites and institutes at NHLS were included in the study. Median NHLS staff levels were collected for each category during June or July (with the exception of 2011, which used November data because the programme was launched in August).

The data from OHASIS were exported to Microsoft Excel 2016 for analysis. The incidence rates of TB per occupational category and per year were calculated. In this study, the employees were divided into occupational groups based on potential exposure to TB patients or specimens as outlined in their job descriptions. The sum of the yearly medians, for each occupational category, provided the estimate of the time-at-risk, in years, that employees contributed to a study. Rate ratios were calculated and a *p* value of ≤ 0.05 was considered significant. 

The study sample does not capture exposures without a confirmed TB diagnosis, or TB diagnoses that were not reported through OHASIS. To encourage reporting, the OHNs followed up on each exposure report until diagnosis was determined and treatment provided if indicated. How reported incidents and factors promoting exposure, once identified, are actually acted upon in the workplace is the subject of a separate analysis. 

Ethical approval to use secondary data from OHASIS was obtained from the University of Witwatersrand Human Research Ethics Committee (Medical), clearance certificate number: M180480 and the University of British Columbia Behavioural Research Ethics Board, certificate number: H10-00360.

## 3. Results

### 3.1. Incidence of Tuberculosis

There were 92 cases of TB reported from 2011 to 2019, however there were limitations noted in the initial uptake of OHASIS: from August to December in 2011, only two cases were reported, and no cases were reported in 2012. To increase reporting and follow-up, four additional OHNs were hired between December 2012 and June 2013. An increase in reporting was observed: eight cases in 2013, eight cases in 2014, 18 cases in 2015, 13 cases in 2016, 19 cases in 2017, 20 cases in 2018, and four cases in the portion of 2019 studied. (Table 1).

The cases were 69.6% female and 30.4% male, which was similar to the NHLS employee distribution in 2019, 68.9% female and 31.0% male. Gender distribution remained comparable throughout the study. Age was not captured for the confirmed TB cases and therefore cannot be compared with the demographics of the workplace as a whole. 

The incidence rates per 100,000 person-years among general workers is 421.85, more than twice as high as the incidence in any other occupational category (Table 2). The rate ratios show that an increase in expected occupational TB exposure was associated with a protective mechanism as all rate ratios are below 1, and statistically significant (*p* < 5, 2-sided). The least at risk group identified were medical staff.

### 3.2. Factors Affecting Incidence of Tuberculosis Cases

The OHN identified factors influencing TB exposure during their investigation of a case, and these factors were arranged into six groups: environment, worker, patient, organizational, equipment, and work practice. Each factor was represented as a percentage of the cases, in the specified occupational category, that indicated that factor type. Medical staff did not indicate any environmental or work practice factors in their identified cases and are excluded from the graphs. Six cases did not indicate any factors, and this was interpreted as an incomplete form and the cases were not included in the comparison. 

For 86 cases, the OHN identified one or more factors that, based on their assessment, contributed to the employee being exposed to TB. These are displayed broadly based on factor category (Figure 1). Variables that describe each of the factor categories are presented in Appendix A. Notable was the lack of diversity in the medical staff occupational category, 50% of all medical staff cases indicated employee factors as an influence and 75% indicated organizational factors. Employee and organizational factors are influential factors across all occupational groups. Patient factors were only selected in general worker cases, and these were the least indicated factors. 

Table 3 shows that organizational factors differ the most in the general worker category, which was the only category to indicate “lack of policy and procedure” and “working alone” as contributors to TB exposure; also, it was the only category not to indicate “lack off personal protective equipment” (PPE).

The most influential work practice factor was “practices not followed”, which represented 75% of the general worker group, followed by “practices unclear”. No medical staff cases indicated workplace practices as a factor contributing to TB transmission.

Environmental factors “ventilation”, “workspace layout” and “other” were represented in groups other than medical staff, which did not indicate any environmental factors. “Construction” and “temperature” were only indicated as factors in the laboratory skilled group (Table 3).

Employee factors varied greatly among occupational categories (Table 3). Of significance is that all medical staff indicated “other”, which may suggest limitations of the options. “Inadequate training” and “pre-existing condition” were influential in the remaining occupations.

Importantly, OHASIS also collected information on the incident causes, self-identified by the TB cases. The categories of causes that were indicated in at least one case include: cleaning, office work, laboratory work, patient interaction, and non-specific (Appendix A). Of those cases that completed this section, the most common self-identified cause of TB infection was laboratory work: medical staff (67% of cases), laboratory skilled (94% of cases), laboratory unskilled (94% of cases) and general workers (53% of cases). “Patient interaction” was common in medical staff (67% of cases) and between 0% and 6% in the remainder of occupational categories. General workers identified “office work”, “routine cleaning” and “non-specific” causes (26%–32%) as contributors to exposure (Table 3). 

Seeking treatment via private physician was most common in medical staff (67% of cases), while others remained comparable; laboratory skilled (46% of cases), laboratory unskilled (50% of cases) and general workers (30% of cases). Being admitted to hospital was nearly as common and similarly distributed through the occupations: medical staff (33% of cases), laboratory skilled (42% of cases), laboratory unskilled (50% of cases) and general workers (45% of cases). The use of on-site first aid was only used in one case, by a laboratory skilled worker. The occupational health unit was never used by medical staff.

## 4. Discussion

South Africa has a high rate of TB in the general population, with many people being unaware of their diagnosis, which means that high quality infection control and surveillance programmes are required for the control and decreased incidence of occupational TB. As prevention and control in high TB incidence LMICs are constrained by limited resources and funding, weak or non-existent occupational health programmes prevail despite the increased risk [7]. Complicating the matter, for the same reasons there is also a lack of surveillance, leading to limited data regarding the extent and transmission of occupational TB infection [7].

TB transmission is pronounced in laboratories and medical wards [16] due to persistent occupational exposure to sputum samples from active TB patients [5,6,17]. This study presents a unique opportunity to observe the insights that can be systematically gained through the collection of information from a workplace occupational health surveillance programme. The surveillance programme was implemented in a predominantly laboratory-based healthcare enterprise based in a LMIC experiencing a high burden of TB. 

Despite having the least contact with specimens and patients, general workers had a significantly higher TB incidence rate than other occupational groups and were the closest to the national incidence rate. The general worker represents employees with the least amount of medical education, lowest comparative average salaries, and the least amount of institutional occupational health training and ongoing support. The gap in knowledge for this group was in the ability to identify risks and protect themselves against disease transmission. Through the use of the surveillance system, specific areas that influence high levels of TB infection in general workers were identified.

“Pre-existing conditions” and “illness” were factors identified through OHASIS as associated with TB infection. This could be associated with HIV infection or other immune-lowering diseases, which are more prevalent in those with less education and lower socio-economic status, thus increasing the likelihood of TB infection [6,18,19,20]. This highlights the well-documented observation that socio-economic status confers considerable risk for TB, and hence it is particularly important to ensure that policies and training are in place to protect these workers in the work environment.

The lack of PPE use was a significant factor for TB infection in all occupational categories except for the general worker. Given the high incidence rate for general workers, there is possibly a lack of knowledge regarding the importance of PPE use as a means of TB prevention. It is also likely that in most institutions, general workers are not fit tested for an N95 respirator or aware of the protective abilities of wearing such a respirator. Despite not being aware of the PPE use, the general worker did recognize that support systems were missing or may not have been aware of existing policies and procedures. This claim was supported by the identification of “lack of training” and “lack of policy or procedure” as significant factors for transmission in this group. “Sample handling” was also identified in a significant number of TB cases in general workers, which may not have been appropriate given their occupational role. This is undoubtedly linked to practices not being followed, as this again shows a lack of knowledge and awareness of disease transmission and the risk of not following procedure. Overall, it is important to recognize that clinical and non-clinical HCWs have different levels of education and occupational health training, which affect their ability to recognize and respond to potential exposures. Medical staff selected “other” most frequently when completing the OHASIS report. This may be because the questionnaire does not capture significant causes or factors for this group. The addition of factors related to TB exposure that closely relate to a clinical approach should be considered. For example, limited patient isolation rooms, patient assessments or TB specific triage are situations that increase potential exposure.

Despite having varied rates for TB, medical response appeared equal, as TB cases were similarly admitted to hospital across all occupational groups, suggesting that employees at NHLS were provided with equitable accessibility to treatment regardless of occupational group. 

There are several factors that must be kept in mind in interpreting the rates presented here. First, while self-reporting through the web system increased reporting, it is not legally obligatory for workers to report their diagnosis. Underreporting is therefore still an influencing factor, with stigma and fear of TB diagnosis still prevalent in the workplace; and, despite efforts to date, people may still choose not to share their potential exposure with their employer due to fear of job loss. Recall bias may also influence an employee’s response, as a reported exposure may not be followed up immediately by the health and safety representative.

As is true in other health-related domains, reliance on good evidence is pivotal to monitoring workplace safety, influencing procedural changes, and estimating the outcome of policy changes. In our study, we observe that integrating a surveillance system within an established occupational health and safety programme can foster operational improvements rooted in evidence and contribute to positive occupational health outcomes. OHASIS has provided increasingly reliable and valuable data outputs as use and support has grown at NHLS. The lessons learned in implementing a computer-based surveillance system across this complex, multi-facility organization based in a middle-income setting should be further explored, especially when considering the feasibility of using information in high risk workplaces with less established occupational health and safety systems—as needs and potential benefits in such circumstances are especially great.

## 5. Conclusions

The highest rate of TB reported in the surveillance system was in general workers rather than skilled and unskilled laboratory workers. This may be due to the concomitant risks conferred by lower socio-economic status, education levels, and occupational health and safety awareness of the general worker compared to the other groups of health care workers. However, there may also be a reporting bias, related to stigma. Nonetheless, the influence of differing occupational health and safety initiatives and baseline education can be observed in the varied expression of contributing factors across the different occupational groups. Specific actions and duties have also been identified as high risk for TB cases. Overall, the information produced by OHASIS illustrated areas that require intervention, such as where to focus additional TB prevention campaigns, and where further monitoring is required. This is highly valuable information because it not only guides targeted policy and procedural changes to maximize benefit, but the system itself is able to function in an environment where resources are limited and infection rates are high. 

## Figures and Tables

**Figure 1 ijerph-17-01462-f001:**
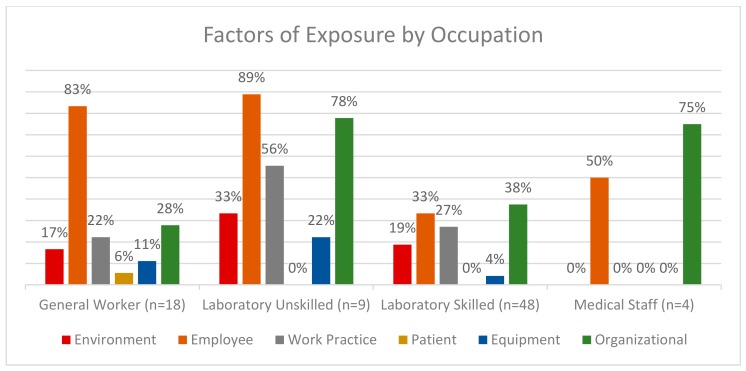
Factors Contributing to TB Exposure by Occupation.

**Table 1 ijerph-17-01462-t001:** Distribution of Cases and Employees.

Variable	2011	2012	2013	2014	2015	2016	2017	2018	2019	Totals
Median # of Employees	5704	5556	5882	5551	5506	5767	5739	5414	5214	50,333
Total TB Cases	2	0	8	8	18	13	19	20	4	92

**Table 2 ijerph-17-01462-t002:** TB Incidence Rates by Occupational Category.

Occupational Category	Cases	Exposed Person-Years	Incidence Per 100,000 Person-Years	Rate Ratio	Rate Difference Per 100,000 Person-Years	*p*-Value
South Africa	-	-	520	-	-	-
General Worker *	20	4741	421.85	-	-	-
Laboratory Unskilled	16	9722	164.58	0.39 (0.20, 0.75)	−257.3 (−459, −55.6)	0.004
Laboratory Skilled	50	31,361	159.43	0.38 (0.23, 0.63)	−262.4 (−452.5, −72.3)	0.000
Medical Staff	6	4508	133.1	0.32 (0.13, 0.79)	-288.8 (−502.1, −75.4)	0.009
Total	92	50,333	182.79	-	-	-

* All groups are compared to the general worker.

**Table 3 ijerph-17-01462-t003:** Factors affecting the incidence of TB cases.

Factors	General Worker N (%)	Laboratory Skilled N (%)	Laboratory Unskilled N (%)	Medical Staff N = (%)	Total N (%)
**Organizational Factors**					
Working alone	1 (17)	0	0	0	1 (2)
Lack of training	2 (33)	8 (30)	3 (36)	1 (25)	14 (34)
Lack of PPE	0	5 (40)	4 (23)	1 (25)	10 (24)
Unsupervised	2 (33)	10 (30)	2 (41)	2 (50)	16 (39)
No policy or procedure	1 (17)	0	0	0	0
**Workplace Practices**					
Practices unclear	1 (25)	5 (36)	1 (20)	0	7 (30)
Practices not followed	3 (75)	4 (29)	3 (60)	0	10 (43)
Extended working hours	0	1 (7)	0	0	1 (4)
Other	0	4 (29)	1 (20)	0	5 (22)
**Environmental Factors**					
Temperature control	0	1 (10)	0	0	1 (6)
Workplace layout	1 (25)	5 (50)	1 (50)	0	7 (44)
Limited space	1 (25)	1 (10)	0	0	2 (13)
Construction	0	1 (10)	0	0	1 (6)
Ventilation	2 (50)	2 (20)	1 (50)	0	5 (31)
Other					
**Employee Factors**					
Inadequate training	8 (50)	9 (45)	4 (31)	0	21 (42)
Inexperience	1 (6)	0	0	0	1 (2)
Illness	0	0	1 (8)	0	1 (2)
Unable to follow instructions	1 (6)	2 (10)	0	0	3 (6)
Pre-existing condition	6 (38)	4 (20)	5 (38)	0	15 (30)
Other	0	5 (25)	3 (23)	1 (100)	9 (18)
**Self-Identified Incident Causes**					
Nonspecific	6 (22)	3 (5)	2 (10)	0	11 (10)
Laboratory work	10 (37)	47 (82)	15 (71)	4 (50)	76 (67)
Office work	5 (19)	1 (2)	2 (10)	0	8 (7)
Cleaning	5 (19)	3 (5)	2 (10)	0	10 (9)
Patient interaction	1 (4)	3 (5)	0	4 (50)	8 (7)
**Medical Response**					
Attended casualty	4 (17)	6 (10)	1 (5)	0	11 (10)
On site first aid	0	1 (2)	0	0	1 (1)
Admitted to hospital	9 (39)	21 (35)	8 (36)	2 (29)	40 (36)
Private physician	6 (26)	23 (38)	8 (36)	4 (57)	41 (37)
Occupational health unit	4 (17)	9 (15)	5 (23)	1 (14)	19 (17)

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
