# Peer review of "Occupational Tuberculosis Among Laboratory Workers in South Africa: Applying a Surveillance System to Strengthen Prevention and Control"

_ijerph, 2020, doi:10.3390/ijerph17051462_

Round 1

Reviewer 1 Report

1) I'm sorry but I think that the "Occupational groups" are not well classified by means of an "exposure score" to TB and consequently the "unexposed" groups are not clear;

2) Is not clear described in the methods the calculation of the person-years; 

In these conditions also the RR are not significant. I think that with major revisions in the exposure assessment, person-years calculation and with a new RR calculation the results could be more interesting. 

Reviewer 2 Report

The manuscript by Garnett et al. presents data on occupational TB cases over the 8-years period in a high TB incident country and factors contributed to TB exposure as self-identified by the cases. An important conclusion is drawn in the study regarding the occupational group (general workers) who have the least contact with TB patients and biomaterial but at the same time have the highest TB incidence rate compared to other occupational groups. These data point out the need for re-considering training programmes for staff in medical institutions, including but not limited to TB labs.

I have a few, mostly minor, comments.   

Abstract

Please use the abbreviation ‘TB’ consistently throughout the abstract

Introduction

You cite studies mostly from South Africa (references 1-6) which is relevant to the manuscript. But are there any studies from other settings? Please consider adding some.

Lines 47-52: are you sure this information has been taken from the WHO TB report (reference 7)?

Methods

Please consider moving the paragraphs starting from line 172 and Table 1 to the Results section.

Results

Line 220: ‘lack of’

I’m not sure if the current presentation of the findings as figures 1-6 is optimal. The number of cases in each group differs (for example, Fig.2: 5-7-19-3 cases in each group). As a result, 53% for ‘unsupervised’ in Laboratory Skilled with n=19 is very different from 67% in the same category in ‘Medical Staff’ with n=3. I would suggest presenting these distributions in all factors as one table. This table may also contain information given in lines 249-255.

Line 295: the abbreviation hasn’t been introduced before.

Lines 297-299: in the system, is it possible to add a comment when ‘other’ is chosen? That would allow the cases to explain what factor they have in mind.

Conclusions

Is implementation of the surveillance system planned in other settings? Are you planning to make the software publicly available?

Round 2

Reviewer 1 Report

The modifications are not the wonderful but sufficient.